# Current Status and Trends in Prophylaxis and Management of Anthrax Disease

**DOI:** 10.3390/pathogens9050370

**Published:** 2020-05-12

**Authors:** Vladimir Savransky, Boris Ionin, Joshua Reece

**Affiliations:** Emergent BioSolutions Inc., 300 Professional Drive, Gaithersburg, MD 20879, USA; ioninb@ebsi.com (B.I.); reecej@ebsi.com (J.R.)

**Keywords:** Animal Rule, anthrax, anthrax therapeutic, anthrax vaccine, antibiotics, *Bacillus anthracis*, post-exposure prophylaxis, prophylaxis

## Abstract

*Bacillus anthracis* has been identified as a potential military and bioterror agent as it is relatively simple to produce, with spores that are highly resilient to degradation in the environment and easily dispersed. These characteristics are important in describing how anthrax could be used as a weapon, but they are also important in understanding and determining appropriate prevention and treatment of anthrax disease. Today, anthrax disease is primarily enzootic and found mostly in the developing world, where it is still associated with considerable mortality and morbidity in humans and livestock. This review article describes the spectrum of disease caused by anthrax and the various prevention and treatment options. Specifically we discuss the following; (1) clinical manifestations of anthrax disease (cutaneous, gastrointestinal, inhalational and intravenous-associated); (2) immunology of the disease; (3) an overview of animal models used in research; (4) the current World Health Organization and U.S. Government guidelines for investigation, management, and prophylaxis; (5) unique regulatory approaches to licensure and approval of anthrax medical countermeasures; (6) the history of vaccination and pre-exposure prophylaxis; (7) post-exposure prophylaxis and disease management; (8) treatment of symptomatic disease through the use of antibiotics and hyperimmune or monoclonal antibody-based antitoxin therapies; and (9) the current landscape of next-generation product candidates under development.

## 1. Introduction

Anthrax, or a pustular disease closely resembling the clinical presentation of anthrax, has been described in classical literature as early as the writings of Virgil [1]. It was the first disease to be definitely attributed to a bacterium (*Bacillus anthracis*) by Robert Koch in 1877 [2]. Although primarily a zoonotic disease contracted from animals and contaminated soil, it has been identified by the Centers for Disease Control and Prevention (CDC) as a Category A potential agent for bioterrorism [3]. *B. anthracis* was probably used as a weapon in both the First and Second World Wars, and was responsible for deaths in the Soviet Union in 1979, in Japan in 1995, and in the United States in 2001 [3].

Anthrax disease is caused by *B. anthracis*, an aerobic or facultatively anaerobic, Gram-positive, endospore-forming, rod-shaped bacterium [4]. The bacilli are large (1.0–1.5 µm by 3–8 µm) non-motile, and occur singly or in short chains [5]. In the environment, *B. anthracis* can exist as dormant spores, which can persist in dry soil for decades and are highly resistant to disinfectants and temperature [6]. The life cycle of *B. anthracis* comprises a vegetative form and dormant spores. Sporulation is a defense mechanism that occurs when the pathogen’s growth cannot be sustained by the environment. These spores are able to survive multiple environmental challenges and serve as a means of survival for transmission between one host and another [7]. After entering a host by the cutaneous, gastrointestinal, inhalational, or intravenous/injectional route, the spores rapidly germinate and produce significant amount of toxins which are disseminated in the bloodstream, causing clinical symptoms ranging from edema and necrosis to sepsis and venous collapse [4]. In the majority of cases of inhalational anthrax, the spores are transported to the regional lymph nodes where they then germinate and multiply into vegetative bacilli [8]. The germination near the entry site or in epithelial/non-lymphoid tissues has been demonstrated in cutaneous disease, but the significance is uncertain [8].

## 2. Anthrax Pathogenesis

### 2.1. Virulence Factors

Two plasmids, pXO1 and pXO2, play the key role in the pathogenicity of *Bacillus anthracis*. They are responsible for the production of anthrax toxins and the formation of a poly-γ-d-glutamic acid (PGA) capsule [4]. The anthrax toxin genes are located on pXO1, and the capsule biosynthetic operon is located on pXO2. The PGA capsule protects the bacilli from phagocytosis and immune surveillance in a manner similar to that of the polysaccharide capsules of other pathogenic bacteria, such as meningococci and pneumococci. As a result, the capsule mediates the invasive stage of the infection [9,10]. 

Independent polypeptide chains—lethal factor (LF), edema factor (EF), and protective antigen (PA)—comprise the anthrax toxin [11]. The LF, EF, and PA encoded by the pXO1 plasmid combine in binary arrangement to form lethal toxin (LT) and edema toxin (ET) [4]. LF is a zinc-dependent metalloprotease that targets mitogen-activated kinase kinases (MAPKKs or MEKs) [7]. LT cleaves the N-terminus of these enzymes and disrupts signaling pathways critical for regulation of cell cycle, proliferation and cellular stress defense [7,12]. A calmodulin-dependent adenylate cyclase EF causes the intense edema associated with anthrax disease. EF affects signaling pathways in the host cell by increasing cyclic adenosine monophosphate (cAMP) levels [7]. PA is responsible for the translocation of LT and ET to host cells by binding to receptors (TEM8 and CMG2), forming a pore within the membrane to deliver the toxin to the cytosol [7]. Over the last few decades, the mode of action of anthrax toxin was extensively studied and currently is well understood [4,11]. In particular, anthrax toxins have been identified as crucial for *B. anthracis* pathogenesis [8]. Indeed, in humans and most animal models, with the exception of murine models, anthrax is a toxin-mediated disease.

### 2.2. Immunology

ET and LT are capable of entering the cytosol of multiple cell types, including immune cells, and disrupting cell signaling [13]. Anthrax toxins prevent an effective immune response being mounted by rendering dendritic cells and T lymphocytes non-responsive [14,15,16]. Initially, the immune response is triggered by recognition of RNA molecules that coat the surface of the sporulated form [17]. In 2017, Choo et al. reported that anthrax spores can induce responses in human host cells by activating macrophages that recognize vegetative *B. anthracis* using two cell surface receptor proteins, TLR7 and TLR8, which recognize RNA molecules that are embedded in the outer layer of the spore [17]. This provokes a distinct set of immune responses in macrophages and stimulates type I interferons immune signaling pathway. The authors suggest that these interferons incapacitate the immune response, allowing infection to progress to later stages [17].

Early work on adaptive immunity of anthrax traditionally focused on serology, based on the hypothesis that protective immune response is generated against PA and LF. Both PA and LF confer active and passive response to bacterial challenge [18]. Ingram et al. reported that, following anthrax infection and antibiotic treatment, strong T cell memory response could be detected years later [18].

### 2.3. Clinical Forms of Anthrax Disease

Humans can be infected by four major routes: inhalational, cutaneous, gastrointestinal [19], and intravenous/injectional [20]. As a naturally occurring disease, primarily in herbivores, anthrax was one of the major causes of mortality in livestock until the development of live attenuated veterinary vaccines in the 1930s [6]. Even with the development of effective veterinary vaccines, anthrax remains enzootic in sub-Saharan Africa, Asia, and the Americas [6,21]. Outbreaks occur worldwide annually in livestock. In the human population, it has been estimated that 20,000 to 100,000 cases occurred in 1958 worldwide [21]. By the end of the 20th century this had declined to ~2000, with most cases occurring in the developing world [5,6,21]. In Europe and the United States, only sporadic cases are reported. In 2016, six laboratory confirmed anthrax cases were reported within European Union/European Economic Area (EU/EEA), five in Romania, and one in Spain [22]. Most human disease manifests as cutaneous anthrax (>95%). People at risk of contracting cutaneous anthrax are butchers, farmers, veterinarians, and those working with animal hide or wool [22,23]. In the EU, intravenous drug users were identified as an additional risk group from a new clinical manifestation of anthrax, i.e., intravenous/injectional anthrax [20].

Although humans are moderately resistant to anthrax, four types of human anthrax disease are now recognized, none of which are contagious [19,20]. Infectious doses are difficult to determine, but in the absence of skin lesions, 50% infective doses (ID_50_) are in the thousands or tens of thousands of spores.

#### 2.3.1. Cutaneous

As stated previously, in humans, cutaneous anthrax accounts for >95% of cases. The cutaneous form of the disease follows entry of spores into the skin through a scrape or cut. Once inside, spores germinate, forming vegetative bacilli that elaborate virulence factors. Satellite cutaneous bullous lesions may develop. In 5% to 20% of untreated cases, dissemination of the infection via circulatory system may occur [2,5,18].

Lesions commonly appear on the face, neck, hands, and arms. The incubation period ranges from 12 h to seven days [2,5,18]. Initially, a small, erythematous macule or papule typically appears, which subsequently turns brown with a ring of erythema and a vesicle. Satellite vesicular lesions may also appear. Several days later, the vesicular fluid becomes blue-black due to hemorrhage and ulcerates leading to a black eschar by Day 5–7. Non-pitting gelatinous edema (malignant edema), together with a black eschar, is very characteristic for cutaneous anthrax. The most frequently experienced other symptoms are headache, malaise, and low-grade fever [5]. Regardless of treatment, resolution is slow [2].

#### 2.3.2. Gastrointestinal

Gastrointestinal (GI) and oropharyngeal anthrax constitute < 5% of anthrax cases. These types of disease usually stem from the ingesting spore-contaminated meat that may be insufficiently cooked. GI anthrax may be characterized by a gastric ulcer, terminal ilium or cecum, and edema and in regional lymphatic vessels [5]. Oropharyngeal anthrax presents with sore throat, ulceration of the oral cavity, lymphadenopathy of the cervical and/or submandibular lymph nodes, and frequent swelling of the neck [5]. Symptoms typically develop between two to five days after exposure, and mortality occurs in ~50% of patients with oropharyngeal disease [2]. Patients with GI anthrax usually present with abdominal pain, anorexia, nausea, and vomiting, including occasional hematemesis. These may be followed by the development of ascites, bloody diarrhea, toxemia, toxic shock, and severe prostration [5].

#### 2.3.3. Inhalational

Inhalational anthrax accounts for < 5% of reported cases [5]. The disease results from the inhaling aerosol particles, ranging in diameter from 1 to 5 µm. Larger particles are removed by pulmonary mucociliary clearance. Animal studies have shown that inhaled spores, engulfed by alveolar phagocytes as well as dendritic cells, are then transported into tracheobronchial and mediastinal lymph nodes, where germination occurs, leading to the formation of the capsule and release of LT and ET [6]. The incubation period typically lasts one to six days post-exposure [24]. The early phase of disease manifests as a flu-like illness, with mild fever, headache, fatigue, and myalgia, as well nonproductive cough and mild discomfort in the chest [5,25,26,27]. A short 1–3-day period of improvement may follow the prodromal symptoms prior to rapid deterioration. The second phase of the disease is characterized by high fever, stridor, dyspnea, cyanosis, and septic shock [25,26,27]. Edema of the chest wall and hemorrhagic meningitis may occur in ~50% of cases [5,28] and hemorrhagic pleural effusions have also been reported [5]. During the anthrax letter bioterrorist attack in the United States in 2001, nausea or vomiting were reported, along with severe headache and fatigue [19,29]. Pleural effusions and a characteristic widened mediastinum are typically visible on radiographs of the chest. However, true pneumonitis is not usually observed. In the later stages of illness, the Gram-positive spore-forming bacilli may be present in blood smears. Without prompt intervention, the disease is 100% fatal. Even with treatment, death may occur in up 95% cases if therapy is initiated more than 48 h after the onset of symptoms [25].

#### 2.3.4. Intravenous/Injectional Anthrax

Intravenous (IV) drug use-associated anthrax presents with atypical, serious localized soft tissue infections accompanied by disproportionate tissue edema, often with less pain than other serious soft tissue infections [30]. Fever and localized injection-related lesions are not necessarily present. In the first documented outbreak of anthrax involving heroin users that occurred in Scotland, many of the seriously ill patients died rapidly and presented with symptoms more closely resembling systemic anthrax infection and toxemia, including hemorrhagic meningitis, multi-organ failure, and bleeding diathesis [30]. Genetic analysis of the outbreak identified the strain as a novel Ba4599 phenotype [31]. Over 30% of patients contracting anthrax through IV drug use die [30]. Table 1 summarizes the major components of the different types of human anthrax.

## 3. Summary of Animal Research

As anthrax is relatively rare in humans, only limited clinical data can be collected. As a result, development of prophylactic or therapeutic countermeasures necessitates the use of animal models. An appropriate animal model has to be predictive of the human disease and must recapitulate the human response to the investigational drug [37,38]. This is recognized by the U.S. FDA in the “Animal Rule”, which allows for product approval based on well-controlled animal studies [37].

In animals, the 50% lethal dose (LD_50_) ranges from <10 spores to >10^7^ in more resistant species when administered parenterally [6]. Precise human infectious doses have not been established and depend on multiple factors, including route of infection as well as general health and underlying medical conditions of the exposed individual. Epidemiological evidence indicates that only a small number of spores are needed to develop cutaneous anthrax when infection occurs through a breach in the skin. In the absence of data of the infectious dose in humans, the LD_50_ established in animal experiments can be used [6].

As *B. anthracis* is not regarded as an invasive organism, generally much higher experimental LD_50_s by inhalational or oral routes are used in susceptible laboratory species (Table 2) [6]. Animal models have been extensively used to replicate the symptoms and lethality of the disease, and to demonstrate that antibodies raised against the PA component of the anthrax toxin confer protection against death due to anthrax disease [4,39,40,41]. The cellular and systemic effects of LT and ET are described above and in detail by Moayeri and Leppla [4].

In choosing an animal model for studying anthrax disease, consideration should be given to the relationships between susceptibility of the species to anthrax toxins and its resistance to infection. Host species with greater resistance to bacteremia may be more sensitive to even low doses of the toxins and vice versa [38,50].

Mice have been extensively used in anthrax research. Their small size, low cost, availability of specialized inbred and knockout strains, as well as immunological reagents, make them a valuable research tool for infectious diseases including anthrax. Inbred strains of mice have been shown to be susceptible to inhalation and parenteral challenge with fully virulent *B. anthracis* strains developing a characteristic acute anthrax disease [38]. Moreover, a murine aerosol challenge model has been developed and successfully used to study anthrax pathogenesis [51]. The course of disease and the resulting pathology observed in complement-deficient A/J mice exposed to aerosolized Sterne strain (pX01+ pX02−) spores mimics that observed in guinea pigs, rabbits, and nonhuman primates (NHPs) exposed to fully virulent strains of *B. anthracis* (e.g., Ames). However, responses to the infection and vaccination differ between mice and humans [38]. 

Rabbits and NHPs are typically viewed as the “gold standard” models of inhalational anthrax, and are widely used for the investigation of new therapeutics and vaccines against the disease [38]. These models are predictive of human anthrax disease because they are highly susceptible to the infection with spores from toxin-producing strains, especially via the inhalational route, and exhibit pathological changes similar to those in humans [38,50]. Rabbit and NHP therapeutic models have also been developed, in which animals are treated on an individual schedule based on clinical signs of the disease which are used as the trigger for treatment [52,53]. These models have and successfully used to evaluate efficacy of antibiotics (levofloxacin) and monoclonal antibodies (obiltoxaximab) [54,55]. However, it is important to note that some animal models may not be suitable for the evaluation of some products. For example, rabbits do not respond well to vaccines containing CpG oligodeoxynucleotide (ODN) adjuvants [56]. As a result, a guinea pig (GP) model has been developed for the assessment of efficacy of anthrax vaccines adjuvanted with CpG ODNs [40]. These are discussed in detail in the 2015 publication by Welkos et al. [38].

## 4. Current Guidelines for Anthrax Management and Prophylaxis

### 4.1. World Health Organization 

In 2008, the World Health Organization (WHO) updated its guidance on the management and prophylaxis of anthrax [6]. *B. anthracis* is susceptible to multiple antibiotics, but early treatment is of paramount importance to eradicate the bacteria before the release of toxins into the circulation. Penicillin G is still widely used in many parts of the world for the treatment of naturally occurring anthrax. The WHO recommends antibiotic therapy and supportive care for gastrointestinal and inhalational anthrax, and for cutaneous anthrax with systemic signs. In severe cases, the WHO recommends a combination of penicillin with a fluoroquinolone (ciprofloxacin or levofloxacin), or a macrolide (clindamycin or clarithromycin). Recommendations for the treatment of gastrointestinal anthrax are a combination of penicillin with an aminoglycoside (streptomycin). In children, penicillin is the antibiotic of choice for treatment. Guidance of the length of treatment is not clear cut. In general, WHO recommends 3 to 7 days therapy for uncomplicated cutaneous anthrax and 10 to 14 days therapy for systemic infections. The WHO also offers guidance on treatment of the more severe forms of the disease, e.g., anthrax meningoencephalitis, treatment during pregnancy and of immune-compromised or -suppressed individuals and surgery as an approach to management of GI anthrax [6]. In addition, WHO recommends isolation and vaccination for livestock in response to an outbreak of disease and prophylactic vaccination in areas where anthrax is enzootic, as well as makes recommendations for responding to anthrax when used in a bioterrorist event, such as the one that occurred in the United States in 2001 [29]. These mirror guidelines published in 2002 [19] and involve prolonged post-exposure treatment of up to 60 days with ciprofloxacin or doxycycline with or without three doses of anthrax vaccine [6].

### 4.2. Centers for Disease Control and Prevention

The CDC recommends prophylaxis with ciprofloxacin (500 mg PO bid) or doxycycline (100 mg PO bid) after inhalational exposure [57]. The 2019 CDC guidance includes recommendations for hemodynamic support and mechanical ventilation, adjunctive corticosteroids, antibiotic prophylaxis and treatment and prevention of the disease with vaccination with AVA at 0, 2, and 4 weeks in individuals who are suspected of having been exposed but do not have active disease. The guidance specifies “*a 42-day antimicrobial regimen when initiated concurrently with the first dose of AVA or for 14 days after the last AVA dose, whichever is later (not to exceed 60 days) for immunocompetent adults as post-exposure prophylaxis*” [58]. Furthermore, the guidance included the addition of an antitoxin to antimicrobial therapy in those patients in whom a systemic involvement is suspected. The CDC has developed a clinical framework for the management of a mass casualty incident illustrated in Figure 1 [34,35].

### 4.3. United States Department of Defense (DOD)

In 2015, the DOD clarified the mandatory requirements and voluntary receipt of anthrax immunizations for DOD personnel and family members. This guidance makes anthrax vaccination mandatory for all personnel and contractors who are assigned to, or are being deployed to, the U.S. Central Command area of responsibility for 15 consecutive days or longer [59].

### 4.4. United States Department of Agriculture (USDA)

The USDA recommends the vaccination of animals as a preventative measure and timely antibiotic treatment post-exposure. As anthrax is a notifiable disease, it must be reported to State authorities [60].

## 5. Regulatory Aspects of Anthrax Medical Countermeasures (MCMs) Development under the U.S. Food and Drug Administration (FDA)

It is neither feasible nor ethical to perform human trials to support the development of MCMs against anthrax. The Animal Rule (21 CFR 314.600-650 for drugs; 21 CFR 601.90-95 for biologics; effective 1 July 2002) allows drug approval and biologics licensure without conducting human efficacy studies if these studies are not ethical and field trials are not feasible. In these cases, marketing approval may be granted based on adequate and well-controlled efficacy studies in animal models of the human disease or condition of interest. It must be noted that safety should be established and evaluated according the preexisting requirements for drugs and biological products. Therefore, the use of adequate animal models of inhalational anthrax that closely mimic human disease and accurately reflect mechanisms of host–pathogen interaction is critical for the development and licensure of prophylactic or therapeutic countermeasures against the disease [38]. As of the time of this writing, the Center for Drug Evaluation and Research (CDER) has approved 13 products and the Center for Biologics Evaluation and Research (CBER) has approved three products under the Animal Rule [61]. 

Licensure of MCMs under the Animal Rule requires bridging of animal and human pharmacokinetic and/or pharmacodynamic data. This Animal Rule requirement necessitates development and validation of appropriate analytical methods that can be used in both clinical and nonclinical studies. In order to further understand the immune response to PA-based anthrax vaccines, a competitive enzyme-linked immunosorbent assay (ELISA) and a competitive toxin-neutralizing antibody (TNA) assay were developed [62]. Although these assays showed that while the antibody responses elicited by PA-based vaccines were similar among humans, non-human primates (NHPs), and rabbits, the competitive assay suggested that humans may have a significant secondary population of immunoglobulin G (IgG) antibodies that bind to partially or incorrectly folded PA. The use of animal models and improved laboratory techniques has been used to evaluate new products and identify potentially more effective dosing regimens. In 2010, the FDA launched a Medical Countermeasures Initiative (MCMi) program [63]. The aim of this program was to improve the capacity in the United States to respond quickly and effectively to chemical, biological, radiological, nuclear (CBRN), and emerging infectious disease threats, including intentional release into the environment of anthrax spores. The urgency to develop effective countermeasures against anthrax was increased after contamination of the U.S. postal system in 2001 with deadly anthrax spores [57].

## 6. Pre-Exposure Prophylaxis (PrEP)—Vaccination Prior to Anthrax Exposure

### 6.1. Historical Overview of Anthrax Vaccine Development

Inhalational anthrax was first described in the mid-1800s amongst mill workers in Bradford, in the Northern wool manufacturing region of England, who were handling imported wool and hair [64]. In the late 1870s, John Henry Bell inoculated mice and rabbits with blood from a fatal case of “woolsorters’ disease” to transmit the disease. All the animals died, and he found *B. anthracis* in their blood. Subsequent sanitary and disinfection procedures reduced the incidence of the disease. At the end of the 19th century, the first vaccine was developed by Louis Pasteur. In 1935, an attenuated live animal vaccine was developed by Max Sterne, which is still in veterinary use [65]. It was not until the 1950s that a human vaccine (a precursor to AVA) was developed and demonstrated to be effective in a clinical study [66]. The use of this vaccine reduced the number of cases of anthrax mill workers from forecasted 13.35 cases to a single case per month. The findings from this field study resulted in the vaccine being made available to all workers employed in the processing of goat hair in the United States [67].

### 6.2. BioThrax Vaccine—The Only FDA-Licensed Anthrax Vaccine in the United States

The FDA licensed BioThrax^®^ (Anthrax Vaccine Adsorbed) (AVA, Emergent BioSolutions) in 1970 for pre-exposure prophylaxis of disease in persons at high risk for occupational exposure to anthrax [57], and in 2015 for post-exposure prophylaxis (PEP) [68]. AVA is prepared from sterile culture filtrates of the toxigenic, non-encapsulated *B. anthracis* V770-NP1-R strain grown under microaerophilic conditions in a chemically defined protein-free medium. The final product is formulated with Alhydrogel^®^ adjuvant. The primary immunogen in AVA is PA [68].

Initially licensed for pre-exposure prophylaxis (PrEP), the first large scale use was for U.S. military personnel deployed in the Gulf War in 1991 [68]. The use of AVA became mandatory in 1998 for forces deemed to be at high risk for exposure to a weaponized anthrax attack [68]. 

In 2002, the CDC implemented the Anthrax Vaccine Research Development Program (AVRP) to study the safety and immunogenicity of AVA. The initial results of the AVRP study on reduced dosing schedules and intramuscular (IM) route of administration were published in 2008 [69]. This led to regulatory approval of a primary series of 5 doses over 18 months and IM administration of the vaccine, and, subsequently, a 3-dose primary series followed by boosters at 12 and 18 months, based on the evidence that the reduced schedule provided sustained protection against inhalational anthrax [69,70,71].

## 7. Post-Exposure Prophylaxis (PEP)—Vaccination or Treatment after Exposure but Prior to Onset of Symptoms

### 7.1. Use of Animal Rule for Approval of Anthrax MCMs

The 2001 anthrax letter attacks in the United States raised the need to consider extending the licensed indications for BioThrax to include PEP. This was achieved under the FDA Animal Rule, and resulted in the first regulatory approval of a vaccine using this regulatory pathway [72,73]. A series of nonclinical and clinical studies were conducted, which (1) evaluated the ability of AVA, administered post-exposure in combination with antibiotics, to increase survival of animals in comparison to antibiotics alone, in the rabbit anthrax PEP model; (2) determined the neutralizing antibody level that protects against disease based on the relationship between probability of survival and the TNA titer, in the rabbit and NHP pre-exposure models; and (3) assess the ability of the vaccinated human subjects to reach the protective antibody titer within the time period after which the course of antibiotics would have been discontinued in a real-life PEP scenario. This approach was based on the methodology described by Burns [74] and the recommendations of the Vaccines and Related Biological Products Advisory Committee (VRBPAC) [75]. Table 3 summarizes the role of pre-exposure prophylaxis (PrEP), PEP, and passive antibody transfer animal models in licensure of vaccines for a PEP indication.

Other products licensed for anthrax PEP under the Animal Rule include Anthrasil^®^ (Anthrax Immune Globulin Intravenous (human), AIGIV), raxibacumab, obiltoxaximab, and antimicrobials such as fluoroquinolones (levofloxacin and ciprofloxacin) and doxycycline [61]. 

### 7.2. Antibiotics

In a study in NHPs, fluoroquinolones (levofloxacin or ciprofloxacin) were administered for 30 days following anthrax challenge [79]. No clinical signs were observed during antibiotic treatment. After cessation of therapy the monkeys were observed for 70 days. Three deaths occurred in the monkeys who received antibiotics post-treatment and were attributed to the germination of residual spores. Nine of the 10 control animals died within 9 days of anthrax exposure. These results were supportive of the utility of fluoroquinolones for PEP.

Following the 2001 anthrax letter attack in the United States, approximately 10,000 individuals were prescribed either ciprofloxacin or doxycycline as prophylaxis for 60 days [73]. One concern is that strains of *B. anthracis* could become resistant to antibiotic therapy with the use of prolonged courses of therapy. A laboratory study of 18 antibiotics showed the ease with which resistance and cross-resistance develops in vitro to multiple classes of antibiotics and illustrates that strains could be easily engineered to be resistant to antibiotics which could then be used as biological weapons [84]. This is one of the major justifications for the need to develop alternative therapies, such as antitoxins.

### 7.3. Post-Exposure Vaccination

The use of anthrax as a biological weapon prompted an IND application for BioThrax vaccine for the PEP indication. The use of SC injection and a 3-dose schedule (0, 2, and 4 weeks) in the PEP application were informed by earlier studies which evaluated the peak immune response [69,71,85]. Administration using the SC route resulted in higher anti-PA IgG geometric mean concentrations (GMCs) at 4 and 8 weeks compared to the IM administration (49.7 and 94.3 µg/mL versus 30.8 and 84.5 µg/mL). In 2010, the Vaccines and Related Biological Products Advisory Committee (VRBPAC) met to determine the licensure pathway for PA-based anthrax vaccines for the PEP indication under the Animal Rule [74,75]. The committee concluded that the PrEP model should be used as licensure-enabling and that the PEP and passive antibody transfer models should be viewed as supportive (see Table 3).

The U.S. Army Research Institute of Infectious Diseases investigated the post-exposure use of antibiotics, in combination with vaccination, in NHPs subjected to a lethal challenge with aerosolized *B. anthracis* spores [81]. On Day 1 post-exposure, animals were treated with penicillin, ciprofloxacin, doxycycline, doxycycline plus AVA, AVA alone, or saline. The antibiotic treatment was continued for 30 days, while the vaccination was administered on days 1 and 15. All antibiotic regimens conferred 100% protection during and upon cessation of therapy (*p* < 0.002 to < 0.02). All the animals receiving doxycycline plus vaccination survived (*p* < 0.0002). The study demonstrated that prolonged antibiotic therapy for 30 days resulted in significant long-term survival after cessation of treatment but complete long-term survival occurred when antibiotics were combined with vaccination. The study also showed that development of an immune response was predictive of resistance to re-challenge.

The immunogenicity and efficacy of AVA administered in combination with antibiotics post *B. anthracis* challenge were confirmed in later PEP studies conducted in rabbits and NHPs [76]. Infected animals, which did not receive antibiotics or vaccine, began to die on day 1; by day 4, all of these animals succumbed to the disease. Control animals treated with antibiotics plus Alhydrogel adjuvant began to die on day 10; ultimately, 77 of these animals (56%) succumbed to inhalational anthrax. In contrast, animals receiving AVA plus antibiotics were fully protected at the 1:16 dilution of the human dose of the vaccine. The increased survival rate was significant (*p* ≤ 0.006) compared with antibiotic treatment alone.

### 7.4. Antibody-Based Therapeutics

#### 7.4.1. Anthrasil AIGIV

Passive transfer models using animals that had received an anthrax challenge and were then administered purified anthrax immune globulin intravenous (AIGIV, Anthrasil) derived from the plasma of individuals who had been vaccinated with BioThrax vaccine were also used to support licensure [82]. Efficacy of AIGIV was demonstrated in multiple studies, in which rabbits treated with levofloxacin alone or in combination with AIGIV at various time points (between 30 and 96 h) post-exposure. When the treatment was initiated within 60 h post-exposure, the majority (over 88%) of treated animals survived in both levofloxacin-only and combination treatment groups. When the treatment was delayed, addition of AIGIV substantially improved survival outcome, although the differences did not reach statistical significance. These findings highlighted the need for early intervention when using AIGIV in a PEP scenario. 

#### 7.4.2. Raxibacumab

Raxibacumab, an IgG1 recombinant monoclonal antibody, binds to anthrax PA and blocks the effects of the toxin. Raxibacumab was approved by the FDA under the Animal Rule in 2012 for treatment of patients with inhalational anthrax in combination with antibiotics and for PEP when alternative therapies are not available or not appropriate [86]. In rabbit and NHP therapeutic intervention models, animals were treated on an individual basis using detection of PA in serum, a significant increase in temperature, or both as a trigger for treatment. Raxibacumab significantly increased survival compared to placebo in both rabbits (44% vs. 0%, *p* = 0.003) and NHPs (64% vs. 0%; *p* < 0.001) [87]. Furthermore, the addition of raxibacumab to levofloxacin resulted in an 82% survival versus 65% survival in animals that received levofloxacin alone, although the difference was not statistically significant [88]. Interestingly, TNA titers were similar for combination therapy and for levofloxacin alone, suggesting that survivors in both groups were able to mount a toxin-neutralizing immune response [88].

#### 7.4.3. Obiltoxaximab

Obiltoxaximab, another monoclonal antibody against *B. anthracis* PA, was developed in parallel with raxibacumab, but did not receive regulatory approval until 2016. It has since been approved for treatment of inhalational anthrax in combination with antibiotics and for PEP in the absence of appropriate alternative therapies [89]. In a time-based post-exposure treatment study, obiltoxaximab protected 80%, 50%, and 43% or rabbits when administered 24, 36, or 48 h post-challenge, respectively [90]. When tested in combination with doxycycline in the rabbit model, obiltoxaximab significantly (*p* = 0.0051) increased survival compared to the antibiotic treatment alone (90% vs. 50%) [91].

## 8. Treatment of Symptomatic Disease

### 8.1. Antibiotics

The recommendations for antibiotic therapy differ according to the severity and site of infection. Table 4 summarizes the recommendations for antibiotic treatment of naturally occurring, intravenous/injectional anthrax, and anthrax when used as a biological weapon.

In addition to the development of resistant strains, antibiotics may have limited use in the treatment of anthrax as they only clear bacteria from the infected host and do not protect against the effect of toxins [4]. Incidence of resistance to β-lactam antibiotics in *B. anthracis* was first reported in 1990 [92] indicating these drugs may not be optimal for the treatment or management of anthrax. Indeed, β-lactamase expression has been reported in virtually all strains of *B. anthracis* [93]. It is inevitable that this type of resistance will continue to rise. Surveillance will be needed to monitor the spread of resistant strains and guide appropriate antibiotic therapy. Fluoroquinolones are included in the guidelines as alternatives to β-lactams or for use as combination therapy [6,22,34,35,58]; however, the ease with which resistance can be generated in the laboratory is a potential limitation to the widespread use of quinolones for both treatment and prophylaxis [93].

### 8.2. Antitoxin Therapy

Two systematic reviews of antitoxin therapy have been published and include animal studies and case reports [94,95,96]. The authors of both reviews conclude that use of antitoxin therapy may be limited by supply and that further studies are needed to justify the resources necessary to maintain antitoxins in national stockpiles.

## 9. Next-Generation Product Candidates

### 9.1. AV7909

The anthrax vaccine candidate AV7909 is being developed as a next generation vaccine for PEP. The vaccine consists of AVA adjuvanted with the immunostimulatory oligodeoxynucleotide (ODN) CPG 7909. Early proof-of-concept data [97,98] suggested enhancement of immune response to AVA by CPG 7909. Subsequent clinical trials and studies in NHP and GP models confirmed that AV7909 induced a protective immune response more rapidly than the licensed AVA vaccine [99,100,101,102], which may be particularly beneficial for post-exposure use of the vaccine. Furthermore, the TNA level associated with a 70% probability of survival in animals was substantially lower than the corresponded threshold established for the AVA [99]. A study in the GP model also demonstrated that AV7909 conferred added survival benefit in a PEP scenario compared to the antibiotic treatment alone [103], thus confirming the potential utility of this vaccine for PEP.

### 9.2. Recombinant Protective Antigen-Based Vaccines

Over the years, substantial efforts have been made to develop a recombinant anthrax vaccine, based on the wild type or mutated recombinant PA (rPA). Although inherent instability of the PA protein [104,105] presents a considerable challenge to the development of rPA-based vaccines, a number of such vaccine candidates, including those based on bacterial and plan expression systems, viral or bacterial vectors, and DNA vaccine technology, are currently under development [105]. Several rPA vaccines have reached a clinical phase of development, demonstrating a favorable safety profile and strong immunogenicity [106,107,108,109,110] as well as strong protective efficacy in relevant animal models [111,112,113,114,115,116,117,118]. Similar to AVA, rPA-based vaccines demonstrated added value for PEP, compared to antibiotics alone in animal post-exposure efficacy studies [78] and human antibodies induced by such vaccines protected animals against anthrax challenge in passive transfer studies [119]. Most importantly, rPA vaccine-induced TNA titers strongly correlated with protection in pre-exposure rabbit, guinea pig, or NHP models [112,120,121,122,123,124,125].

### 9.3. Novel Antibiotics

Development of novel antibiotics for treatment of anthrax has also been actively pursued in recent years. In particular, daptomycin [126] and omadacycline [127] have shown promise as therapeutics against anthrax disease. The former is a lipopeptide produces by the soil bacterium *Streptomyces roseosporus*, which, unlike other lipopeptides that are active against Gram-positive bacteria, acts via a unique mechanism, disrupting the bacterial cell membrane [128]. The latter is a broad-spectrum, semisynthetic, ribosome-binding inhibitor of protein synthesis, which is a member of the aminometholcycline family of tetracycline derivatives designed to overcome the known tetracycline resistance mechanisms, such as efflux and ribosmal protection [129]. Both were comparable to ciprofloxacin in mouse inhalation challenge studies [127,130]. Other promising candidates include a fourth-generation flouoroquinolone, gatifloxacin, and a semisynthetic lipoglocopeptide dalbavancin, both of which demonstrated the ability to protect mice against inhalational *B. anthracis* challenge when administered post-exposure [130].

## 10. Conclusions

Anthrax has been described for over two millennia [1]. To this day, and despite the introduction of vaccines as a preventative measure, it remains a significant health concern in both humans and livestock in many parts of the developing world where the presentation of the disease is primarily cutaneous. The use of *B. anthracis* as a biological weapon in the twentieth century and as a potential agent of bioterrorism in the early twenty-first century intensified the need to develop effective countermeasures to use in case of an attack [131]. The focus of this development has been on the prevention of inhalational anthrax caused by *B. anthracis* spores.

The ability of the bacteria to sporulate and survive multiple environmental challenges has focused the development of therapeutic agents that are effective for PEP. The relative rarity of anthrax in humans means that investigational therapeutics and prophylactic countermeasures must utilize animal models that closely replicate both human disease and host–pathogen response. In general, the preferred investigational models are rabbit and NHP [38]. A GP model has been developed as an alternative approach for the investigation of CPG 7909-adjuvanted AVA [38,99].

In terms of the treatment of anthrax, antibiotic therapy is currently recommended (Table 4). However, the potential for germination of residual spores after cessation of antimicrobial therapy leading to clinical relapse necessitates prolonged and/or combination antibiotic treatment that is marked by poor adherence to the treatment regimen [132]. Furthermore, antibiotics directly cannot counteract the effects of anthrax toxins, and their utility may be limited by the potential for the development of antibiotic-resistant strains. Additionally, antibiotic-resistant strains could be developed in the laboratory setting and deliberately released into the environment, further highlighting the threat of *B. anthracis* as a biological weapon.

Newer approaches to the management of anthrax include antibody-based therapies and antitoxins as both therapies and prophylactic agents. Although adding to their potential use for management of the disease and PEP, these agents are limited both by cost and difficulty of mass production. Other approaches include those aimed at reducing the time to protective immunity, single-dose vaccination schedules, and investigation of alternative adjunctive therapies [133]. Further investigation and studies of these agents are needed to assess their clinical and prophylactic utility.

## Figures and Tables

**Figure 1 pathogens-09-00370-f001:**
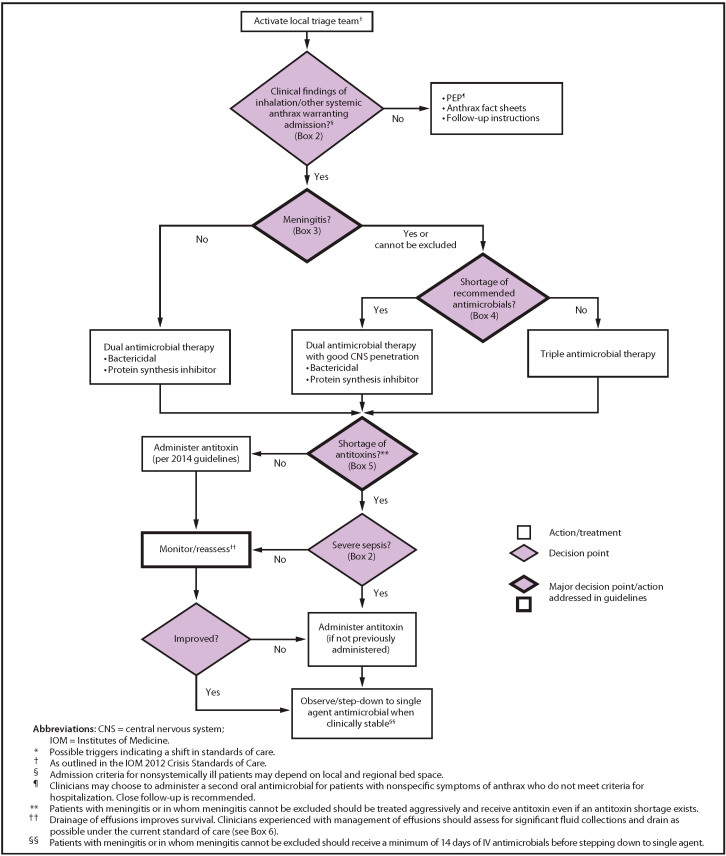
Clinical framework and medical countermeasure use during an anthrax mass casualty incident. Source: CDC, available on CDC website at no charge [34].

**Table 1 pathogens-09-00370-t001:** Comparison of cutaneous, gastrointestinal, inhalational and intravenous/injectional anthrax.

	Cutaneous Anthrax	Gastrointestinal Anthrax	Inhalational Anthrax	Intravenous/Injectional Anthrax *
References	[32]	[6,33]	[27,34,35]	[30,36]
Occurrence	Endemic areas and middle-income countries	Consumption of undercooked meat in endemic areas	Bioterrorism, bioweapon, sporadic cases in wool handlers, drummers, drum-makers and persons exposed to infected animals	Drug users, industrial countries
Incubation period	1–17 days	2–5 days	1–6 days; periods up to 43 days have been observed	1–10 days
Lesion site	Exposure site, mostly superficial	Abdominal pain, vomiting (including hematemesis), hematochezia, and occasional watery diarrhea	Hemorrhagic lymphadenitis, widened mediastinum, meningeal edema and hemorrhage, pleural effusions, pulmonary edema, and hemorrhagic meningitis	Injection site, soft tissue infection with necrosis
Severity of infection	Mild to severe, rarely complicated, up to 20% die if not treated	Early diagnosis is difficult, resulting in high mortality	Severe	Severe
Diagnosis	Patient history, patient examination, laboratory results	Patient history, patient examination, laboratory results	Early diagnosis is difficult. Patient history, examination, microhemagglutination test specific for PA, X-ray, microbiology (blood, sputum)	Patient history, patient examination, laboratory results
Treatment	Antibiotics, supportive care in severe cases	Antibiotics, supportive care	Antibiotics and supportive care, immunoglobulins	Supportive care, of antibiotic combination treatment
Duration of antibiotic treatment	3–5 days	3–5 days	≥2–3 weeks, then 60 days from onset of illness	10–14 days, with up to 60 days for intranasal drug users
Surgical intervention	Rarely	Often due to ascites and/or peritonitis	Rarely	Debridement, reconstructive surgery may be required
Mortality	<1%	4–50%	85–90%. With aggressive treatment, mortality can be reduced to 45%	>30%

* Includes IV and injectional drug misuse.

**Table 2 pathogens-09-00370-t002:** Susceptibility of laboratory animals to *B. anthracis*.

Species	LD_50_ Parenteral Route	LD_50_ Inhalational Route	Primary Pathophysiological Factor(s)	References
Mouse	< 10–151	14,500	Bacteremia	[42,43]
Guinea Pig	< 10–50	16,650–40,000	Bacteremia, Toxemia	[40,44]
Rat	10^6^	-	Toxemia (resistant to infection)	[43]
Hamster	10	-	Bacteremia	[45]
Rabbit	-	105,000	Toxemia	[46]
Cynomolgus Macaque	-	34,000–110,000	Toxemia	[47]
Rhesus Macaque	-	30,000–172,000	Toxemia	[48]
African Green Monkey	-	10,000	Toxemia	[49]

**Table 3 pathogens-09-00370-t003:** The role of four key animal models used in the regulatory pathway for obtaining a PEP indication.

Model	Objective	Role in Establishing TNA Threshold of Protection	Species	References
Pre-exposure Prophylaxis (PrEP)	Demonstrate correlation between pre-challenge TNA levels and probability of survival	Appropriate: • Immune response induced solely by vaccination • Protection conferred by both circulating TNA and immune memory • Animals challenged at the time point that approximates when residual spores may germinate	NZW Rabbit	[76,77]
			Rhesus and Cynomolgus Macaque	[70,76]
Post-exposure Prophylaxis (PEP)	Demonstrate added benefit of vaccine compared to antibiotic treatment alone, in a post-exposure setting	Limited: • Dynamic model of rapidly progressing disease • Immune response results from both vaccination and infection	NZW Rabbit	[76,78]
			Rhesus Macaque	[79,80,81]
Passive Transfer	Demonstrate that neutralizing antibody alone is capable of protection	Limited: • Protection conferred by circulating antibody only • Demonstrates that antibody can protect, but overestimates protective levels of antibody compared with active vaccination	NZW Rabbit AIGIV administered at onset of clinical signs of disease	[82,83]
			NZW Rabbit AIGIV administered as timed post-exposure prophylaxis	[82,83]
			Cynomolgus Macaque AIGIV administered at onset of clinical signs of disease	[83]

AIGIV: anthrax immune globulin intravenous. Adapted from Longstreth et al. 2016 [73].

**Table 4 pathogens-09-00370-t004:** Suggested antibiotic therapy for anthrax.

Category	Antibiotic	Duration
Naturally occurring anthrax	First choice *: • Procaine penicillin G, 0.6–1.2 M units IM q 12–24 h Penicillin G, sodium or potassium 4 M units IV q 4–6 h • Amoxicillin 500 mg PO q 6–8 h Alternative *: • Doxycycline 100 mg IV/PO q 12 h Ciprofloxacin 200–400 mg IV q12 h, followed by 500–750 mg PO q12 h	3–5 days (up to 3–7 days) for cutaneous anthrax without complications; 10–14 days for systemic anthrax ^†^
Intravenous/injectional anthrax	Combination of antibiotics, plus surgical debridement, followed by reconstructive surgery if required	10–14 days, with up to 60 days for intranasal drug users
Biological weapon or bio-terrorism-related anthrax	• Ciprofloxacin 200–400 mg IV q 12 h, followed by 500–750 mg PO q12 h • Doxycycline 100 mg IV/PO q12 h	42–60 days

* For mild cutaneous anthrax, antibiotics may be administered orally. For severe cutaneous or systemic anthrax, intravenous antibiotics must be administered initially; therapy may be changed to oral once body temperature has returned to normal. ^†^ In cases of disseminated infection, the antibiotic selected initially must be combined with one or two of the following; penicillin, ampicillin, ciprofloxacin, imipenem, meropenem, vancomycin, rifampicin, clindamycin, linezolid, streptomycin, or another aminoglycoside. If the patient presents with meningitis, a combination of at least two antibiotics with the ability to penetrate cerebrospinal fluid must be administered. In addition to antibiotics, an antitoxin may also be administered given, if available. Source: Doganay, 2017 [2], and Bower, 2015 [34].

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
