# Peer review of "Current Status and Trends in Prophylaxis and Management of Anthrax Disease"

_pathogens, 2020, doi:10.3390/pathogens9050370_

Round 1

Reviewer 1 Report

The review manuscript by Savransky et al. discusses the details of anthrax disease, prophylaxis, available and potential treatments, and vaccine development. Although the manuscript is written smoothly and there are a lot of topics covered, there is no focus in the review overall. The authors discuss several details and it reads like an anthrax manual, rather than a review article. In general, the problem lies with the novelty of the topic. Most of the information provided is already there in the literature years ago. History of disease and epidemiology has been discussed several times. Another problem is that the authors have not cited several relevant and recent references. My specific comments are as below:

  1. The authors need to highlight the unique features of this article. They need to tell the missing information in the literature and what this article contributes.
  2. The authors have not mentioned at most places the importance of bacillus spores. It is critical because the disease spread and biothreat are because of spore stability and dissemination.
  3. The abstract is not comprehensive. The first few sentences do not match with the rest of the details.
  4. Several important references are missing when there is a discussion about the animal models of anthrax. Authors need to cite these- Arora et al, Curr Topics in Medicinal Chem 2017; Henning et al, Clinical and vaccine immunology, 2012; Loving et al, Infection and immunity, 2007. 
  5. Introduction line 47, "Bacillus anthracis pathogenicity...", this sentence is incomplete/need attention.
  6. Introduction, line 32, there is a typographical error in bacilli size.
  7. Line 125 onwards, Section- Inhalational anthrax, this section is vaguely written with a lot of errors. Please reframe.
  8. Sections 3.2 and 4.3 both have the same topic of the BioThrax vaccine. 

Reviewer 2 Report

Comment 1: The introduction includes too many sections; it should end with line 46.  The sections originally in the introduction should be renumbered.

Comment 2: Lines 47-63 should be a section entitled Anthrax Pathogenesis.

Comment 3: For the statement on line 68, reference 17 would be more appropriate instead of reference 12.

Comment 4: The second paragraph of section 1.2 does not belong in this section.  It is less about pathophysiology and more about epidemiology.

Comment 5: In Table 1, there is no duration listed for gastrointestinal anthrax.

Comment 6: It seems the wrong reference was used in line 163 (reference 38).

Comment 7: In Table 2, reference 45 was used for the rat species; however, this reference details anthrax challenge of guinea pigs.

Comment 8: In line 187, references 38, 44, and 45 should not be included here since they pertain to animal models other than rabbits and NHPs.

Comment 9: The paragraph beginning at line 192 is redundant.  This information was just discussed in the preceding paragraph.

Comment 10: Section headings 3.2.1 and 3.2.2 are not necessary.  The information can be included under section 3.2.

Comment 11: The Table 3 heading is not inclusive of all the models presented in the table.

Comment 12: In Table 4, no duration is listed for the third category (Biological weapon or bioterrorism-related anthrax).

Comment 13: Section 5.1.1 should be 5.2.

Comment 14: In section 6, more information (e.g. review of studies) should be provided for the next-generation product candidates.

Comment 15: Much of the information in the section 7 (Conclusions) regarding B-lactamase resistance belongs in section 5.1.

Comment 16: Some statements in section 7 are simply repeats of statements made earlier in the review.

Round 2

Reviewer 1 Report

Overall I am satisfied with the changes made by the authors and impressed by their literature review. In the current version, the information is comprehensive and will be an important contribution to the field. My only comment is about the resolution of figure 1. The figure needs to be prepared with better quality. It is not readable in the current format.

Reviewer 2 Report

1. The paragraphs beginning on lines 35 and 40 should be one paragraph.

2. The paragraph beginning on line 70 should not be a separate paragraph.

3. The phrase "...and anthrax is not considered a contagious disease" beginning on line 106 should be moved elsewhere or made as a separate sentence.  Possibly, it can to moved to the end of the sentence beginning on line 104. It would read "Although humans are moderately resistant to anthrax, four types of human anthrax disease are now recognized, none of which are contagious."

4. With inhalation anthrax, spores are engulfed by dendritic cells as well, not just alveolar macrophages (line 134).

5. The sentence beginning on line 165 ("In order to develop...") is redundant.  The preceding sentence basically states the same point.

6. A new paragraph should not be started on line 230.

7. The text in Figure 1 is blurred.

8. The Table 3 heading only mentions PEP; however, the table includes PrEP and passive antibody transfer as well.
